# Deducing genotypes for loci of interest from SNP array data via haplotype sharing, demonstrated for apple and cherry

Alexander Schaller¤a, Stijn Vanderzande¤b, Cameron Peace*

Department of Horticulture, Washington State University, Pullman, WA, United States of America

¤a Current address: Department of Environmental Horticulture, University of Florida, Gainesville, FL, United States of America
¤b Current address: Plant Breeding, Wageningen University and Research, Wageningen, The Netherlands
* cpeace@wsu.edu

**Data Availability Statement:** All relevant data are within the manuscript and its Supporting Information files.

## Abstract

Breeders, collection curators, and other germplasm users require genetic information, both genome-wide and locus-specific, to effectively manage their genetically diverse plant material. SNP arrays have become the preferred platform to provide genome-wide genetic profiles for elite germplasm and could also provide locus-specific genotypic information. However, genotypic information for loci of interest such as those within PCR-based DNA fingerprinting panels and trait-predictive DNA tests is not readily extracted from SNP array data, thus creating a disconnect between historic and new data sets. This study aimed to establish a method for deducing genotypes at loci of interest from their associated SNP haplotypes, demonstrated for two fruit crops and three locus types: quantitative trait loci $Ma$ and $Ma3$ for acidity in apple, apple fingerprinting microsatellite marker GD12, and Mendelian trait locus $R_f$ for sweet cherry fruit color. Using phased data from an apple 8K SNP array and sweet cherry 6K SNP array, unique haplotypes spanning each target locus were associated with alleles of important breeding parents. These haplotypes were compared via identity-by-descent (IBD) or identity-by-state (IBS) to haplotypes present in germplasm important to U. S. apple and cherry breeding programs to deduce target locus alleles in this germplasm. While IBD segments were confidently tracked through pedigrees, confidence in allele identity among IBS segments used a shared length threshold. At least one allele per locus was deduced for 64–93% of the 181 individuals. Successful validation compared deduced $R_f$ and GD12 genotypes with reported and newly obtained genotypes. Our approach can efficiently merge and expand genotypic data sets, deducing missing data and identifying errors, and is appropriate for any crop with SNP array data and historic genotypic data sets, especially where linkage disequilibrium is high. Locus-specific genotypic information extracted from genome-wide SNP data is expected to enhance confidence in management of genetic resources.

**Funding:** This study was supported by USDA's National Institute of Food and Agriculture-Specialty Crop Research Initiative project "RosBREED: Combining disease resistance and horticultural quality in new rosaceous cultivars" (2014-51181-22378) and the USDA National Institute of Food Agriculture Hatch project 1014919, Crop Improvement and Sustainable Production Systems (WSU reference 00011) for CP, SV, and AS. The funders had no role in study design, data collection and analysis, decision to publish, or preparation of the manuscript.

**Competing interests:** The authors have declared that no competing interests exist.

# Introduction

Accurate genotypic information on identity, parentage, ancestry, breeding value, and performance potential informs effective germplasm management and use [1]. Historically, fruit breeders and collection curators have relied on meticulous passport and crossing records to be confident about identity, parentage, and ancestry and relied on phenotypic data to estimate genetic potential. Increasingly, locus-specific DNA tests for key traits, often based on simple PCR markers, have been used to determine the genotypes (i.e., allelic combinations) at trait loci of interest for cultivars and selections (e.g., [2–5]). In addition, small panels of neutral genetic markers have routinely been employed by germplasm managers to identify duplicates, infer pedigree relationships among germplasm individuals (mostly parent-child relationships), and to calculate overall relatedness among germplasm individuals. (e.g., [6–10]).

Single nucleotide polymorphisms (SNPs) have rapidly become the genetic marker of choice and are replacing previously developed marker types for a given organism. SNP arrays characterizing thousands of loci across the genome have been developed for fruit crops to provide desired genotypic information genome-wide [1, 11–22]. SNP arrays have been used to determine general relatedness among individuals as well as identify specific pedigree relationships [23–27]. SNP arrays have also been used to make genome-wide predictions for apple, cherry, and peach, in which breeding value and performance potential were based on cumulative information from small-effect alleles across the genome and a few large-effect alleles of quantitative trait loci (QTLs) [28–31]. In the RosBREED project [22, 32, 33], SNP arrays were developed and used in apple, cherry, and peach on large breeding germplasm sets that were pedigree-connected and included many important breeding parents and their ancestors [34] to identify and dissect loci influencing fruit quality and disease resistance traits and identify favorable and unfavorable alleles and their associated SNPs [35–45]. The data obtained from these SNP arrays were curated, which included combining SNPs into haploblocks delimited by historic recombination events and establishing the set of observed multi-SNP haplotypes at each haploblock for all genotyped germplasm individuals [46].

SNP arrays are useful new tools but for their routine use in germplasm management and breeding of fruit crops their information needs to be compatible with that of other assays. To characterize their germplasm genotypically, breeders and germplasm users have relied historically on locus-specific assays such as fingerprinting panels of neutral markers and DNA tests that target QTLs and Mendelian trait loci (MTLs). More recently, SNP arrays have become a cost-efficient tool to characterize an individual's genetic composition genome-wide and have become the tool of choice for named and clonally replicated individuals, such as cultivars, selections, parents, and germplasm collection accessions. However, their genotypic data are not readily compared to historic reported genotypic data that breeders and germplasm users already have access to. Furthermore, breeders still rely on locus-specific assays involving DNA markers such as SSRs, SCARs, or single SNPs for large sets of seedlings at early stages of selection because of the genotyping cost per individual. Thus, methods to translate between the outcomes of SNP arrays and locus-specific tests are needed to integrate new and historic data and to integrate data across various germplasm levels. Without such a means of genotypic data alignment that obtains locus-specific information on named and clonally replicated individuals, germplasm users must either run locus-specific markers in addition to SNP arrays, which is an inefficient use of limited resources, or risk losing previous investments that characterized their material.

Various types of locus-specific information exist that would be valuable to extract from SNP array data. Examples of QTLs of interest are the *Ma* and *Ma3* QTLs reported to influence fruit acidity in apple, explaining 66% of phenotypic variance among breeding germplasm derived from nine important apple breeding parents [40]. An example of a MTL of interest is

$R_f$, reported to underlie fruit color in sweet cherry, with two functional alleles that determine the major market classes of "mahogany" and "blush" [4]. Genotypic knowledge of such QTLs and MTLs would provide insight into an individual's performance potential and inform about its potential contribution to the next generation as a parent. An example of a neutral marker used for understanding germplasm relatedness is *GD12* in apple, which is a component of a multi-SSR fingerprinting panel recommended for this crop by the European Cooperative Programme for Plant Genetic Resources *Malus/Pyrus* working group [47], commonly used for studies of apple germplasm relatedness [6, 48].

Therefore, if germplasm users could readily determine for any SNP array-genotyped individual its relatedness-revealing or functional (trait-influencing) alleles at loci of interest, they would be able to utilize their germplasm with increased confidence as well as merge informative data sets that are incompatible currently. Consequently, the objective of this study was to develop and validate a method to readily deduce alleles for any locus using genome-wide SNP array data and demonstrate it in apple and sweet cherry.

## Materials and methods

### Data set

This study involved 121 apple and 60 cherry cultivars and their previously obtained genome-wide SNP data. A wide assortment of apple germplasm forming the RosBREED apple Crop Reference Set was previously assembled [34] and genotyped using the 8K SNP array [11]. In sweet cherry, a Crop Reference Set was also previously assembled [34], and the Breeding Pedigree Set of additional germplasm to specifically represent the Pacific Northwest Sweet Cherry Breeding Program [49], was also included. This cherry germplasm was genotyped using the 6K SNP array [12]. For both crops, the SNP data was quality-checked, phased, and haploblocked to result in two parental haplotypes for each individual in discrete units across each chromosome [46]. Only data for the chromosomes containing the target loci were used in this study. For apple, 247, 129, and 226 SNPs in 59, 53, and 55 haploblocks (HBs) covering chromosomes 3, 8, and 16 were included, respectively. For sweet cherry, 191 SNPs in 26 haploblocks covering chromosome 3 were included (S1 Table).

### Haploblock positions of loci targeted

Genomic positions of the QTLs of *Ma* and *Ma3* in apple, the MTL $R_f$ in sweet cherry, and the SSR locus GD12 in apple were determined in relation to reported haploblocks [46] by identifying the physical position of each locus in the appropriate reference genome and comparing this physical position to those of SNPs in the 8K apple [11, 46] and 6K cherry array [34] and the SNPs' associated haploblocks. For *Ma*, the reported genomic position of the marker [50, 51] was used, and its physical position was determined on the GDDH13 v1.1 apple whole genome sequence [54] accessed via the Genome Database for Rosaceae [52] using a BLAST search [53]. For *Ma3*, the physical location of the informative SNP identified in [43] on the GDDH13 v1.1 apple whole genome sequence [54] was used as the location of the locus. The physical position in the sweet cherry genome [55] of the marker Pav-$R_f$-SSR [4] was used for the $R_f$ locus. The physical position of *GD12* was determined by a BLAST search [53] of the SSR primer sequences against the GDDH13 v1.1 apple whole genome sequence [54] accessed via the Genome Database for Rosaceae [52].

### Allele assignment and haplotype sharing via IBD or IBS

Reported genotypes of cultivars and their ancestors were assembled for the QTLs, MTL, and SSR (Table 1). For *Ma* and *Ma3*, functional alleles of nine important breeding parents,

**Table 1. Reference panel individuals and their locus genotypes used to assign historically reported alleles to SNP haplotype patterns.**

| Apple—*Ma* | | | Apple—*Ma3* | | |
|---|---|---|---|---|---|
| **Cultivar** | **Allele1** | **Allele2** | **Cultivar** | **Allele1** | **Allele2** |
| Arlet | BF2/J-Ma | H-ma | Arlet | $E$-*Ma3* | $C_{ES}$-*Ma3* |
| Aurora Golden Gala | BWs-Ma | I-ma | Aurora Golden Gala | $F$-*ma3* | $G$-*ma3* |
| Cripps Pink | C-Ma | H-ma | Cripps Pink | $A$-*Ma3* | $F$-*ma3* |
| Delicious | BWs-Ma | I-ma | Delicious | $G$-*ma3* | $H$-*ma3* |
| Enterprise | BF2/J-Ma | J-ma | Enterprise | $B_{Mc}$-*Ma3* | $I$-*ma3* |
| Honeycrisp | A-Ma | D-Ma | Honeycrisp | $B_{GG+DO}$-*ma3* | $J$-*ma3* |
| Splendour | H-ma | BWs-Ma | Splendour | $F$-*ma3* | $G$-*ma3* |
| WA 5 | E-Ma | H-ma | WA 5 | $C_{F2}$-*Ma3* | $F$-*ma3* |
| W1 | F-Ma | G-Ma | W1 | $D$-*Ma3* | $F$-*ma3* |

| Apple—*GD12* | | | Cherry—*Rf* | | |
|---|---|---|---|---|---|
| **Cultivar** | **Allele1** | **Allele2** | **Cultivar** | **Allele1** | **Allele2** |
| Beauty of Bath | 155 | 157 | Ambrunes | H1 | H14 |
| Ben Davis | 155 | 159 | Bertiolle | H1 | H14 |
| Braeburn | 159 | 187 | Black Republican | H2 | H9 |
| Cox's Orange Pippin | 155 | 159 | Blackheart | H13 | |
| Delicious | 153 | 159 | Cristobalina | H4 | H14 |
| Duchess of Oldenburg | 161 | | Dzherlo | H6 | H8/17 |
| Esopus Spitzenburg | 155 | 157 | Early Burlat | H11 | H8/17 |
| Golden Delicious | 155 | 195 | Emperor Francis | H1 | H5 |
| Granny Smith | 155 | 159 | Empress Eugenie | H1 | |
| Lady William | 155 | 155 | MIM 17 | H7 | H8/17 |
| Malinda | 155 | 159 | MIM 23 | H7 | H8/17 |
| McIntosh | 155 | 187 | Napoleon | H1 | H2 |
| Montgomery | 157 | 187 | PMR-1 | H13 | H8/17 |
| Northern Spy | 155 | 155 | Schmidt | H1 | H10 |
| Rome Beauty | 157 | | Schneiders | H3 | |
| Russian Seedling | 157 | 161 | Summit | H1 | H10 |
| Wagener | 155 | 155 | | | |
| Winesap | 153 | 159 | | | |
| Worcester Pearmain | 155 | 155 | | | |
| Yellow Transparent | 155 | 157 | | | |

representing a reference panel, were obtained from their previous allocations in [40, 43] (Tables 1 and 2). For $R_f$, functional alleles for a reference panel of 16 pedigree-connected cultivars were obtained from [4] (Table 1, S3 Table). These functional haplotype designations from [4] were used as historically recorded alleles to be deduced here using SNP haplotype data from [46]. For *GD12*, functional alleles for a reference panel of 20 cultivars included in the germplasm set of [34] were obtained from GRIN-Global (www.ars-grin.gov) (Table 1, S3 Table). For each locus, reported alleles were then associated with the haplotypes of the haploblocks they were located within ($R_f$) or with the combined haplotypes of the two haploblocks they were located between (*Ma*, *Ma3*, and *GD12*). "Haplotype pattern" hereafter refers to such single or combined haplotypes associated with specific locus alleles. To conduct these historic alleles-haplotype associations, alleles of homozygous individuals were assigned first. Next, alleles of heterozygous individuals were assigned by comparing extended haplotype patterns to the other individuals. In cases where two individuals shared a locus allele and a flanking haplotype pattern, that locus allele was assigned to that flanking haplotype pattern, which then

**Table 2. Alleles deduced at the acidity trait loci Ma and Ma3 for apple cultivars and selections by haplotype sharing.**

| Allele (ancestral source) | Cultivars with shared haplotypes at the locus (length of extended shared haplotypes in cM) |
|---|---|
| $A$-$Ma$ (Duchess of Oldenburg) | **Duchess of Oldenburg**\*, <u>**Honeycrisp**</u> (68.2) |
| $B_{F2/J}$-$Ma$ (UP_Jonathan) | *Jonathan* (68.2), <u>**Enterprise**</u> (39.3), F$_2$26829-2-2 (32.9) *Idared*, <u>**Arlet**</u>, **Fiesta** (28.4), **Empress**, **Jonamac** (27.5), **Fireside**, **Minnewashta** (25), PRI 14–126 (22.1), Co-op 15, **Topaz** (17.9), PRI 1661–2 (14.9) **Akane**, **Delorgue**, **Sansa** (13.4) |
| $B_{Ws}$-$Ma$ (Winesap) | **Winesap**\* **(68.2)**, <u>**Aurora Golden Gala**</u>, <u>**Delicious**</u>, <u>**Splendour**</u>, **BC 8S-27-43**, **Braeburn**, **Nicola**, **Scired**, **Sonya**, **WA 2** (39.3), **Fuji** (23.9), Montgomery (20.8), **NY 88** (17.9), Cox's Orange Pippin, Elstar, Ingrid Marie, James Grieve, Kidd's Orange Red, Lord Lambourne (12.0), **Empire** (9.4) |
| $C$-$Ma$ (UP_Lady Williams) | *Lady Williams* (68.2), <u>**Cripps Pink**</u> (17.9) |
| $D$-$Ma$ (Frostbite) | **Frostbite**\*, *Keepsake*, **Sweet 16** (68.2), <u>**Honeycrisp**</u> (22.1), M. Floribunda 821 (11.2) |
| $E$-$Ma$ (NJ 27) | <u>**WA 5**</u>, **Co-op 15** (68.2) |
| $F$-$Ma$ (NJ 136055) | *NJ 90* (68.2), <u>**WA 1**</u> (9.4) |
| $G$-$Ma$ (Grimes Golden) | **Grimes Golden**\*, *Goldrush*, *Golden Delicious*, **Ambrosia**, **Blushing Golden**, **Gala**, **Ginger Gold**, **Cripps Red**, **NY 752** (68.2), **Scifresh** (65.7), **PRI 14–126** (59.2), <u>**WA 1**</u> (37.5), **Autumn Crisp** (20.8), **Chinook** (14.9) |
| $H$-$ma$ (UP_Golden Delicious) | *Golden Delicious*, <u>**Delblush**</u>, **Honeygold**, **Pinova** (68.2), <u>**Arlet**</u> (59.2), **Sunrise** (41.9), **Prima** (31.7), Wealthy, Fireside (24.9), <u>**Splendour**</u>, <u>**WA 5**</u>, **Chinook, Sciros** (24.0), <u>**Cripps Pink**</u> (22.1), **Jonafree**, **Tsugaru**, Esopus Spitzenburg, Jonathan, Akane, Monroe, NY 88, NY 752, Northern Spy, Keepsake, Sweet 16, Worcester Pearmain, Fortune (14.9), **Elstar** (13.4) |
| $I$-$ma$ (UP_Delicious) | **Delicious**, *Gala*, *Kidd's Orange Red*, **BC 8S-27-43**, **NY 543**, **Sansa**, **Scired**, **Sonya**, **Spartan**, **WA 2** (68.2), **Nicola** (54.1), **Ambrosia** (20), <u>**Aurora Golden Gala**</u> (14.9) |
| $J$-$ma$ (McIntosh) | **McIntosh**\*, **Macoun**, **Regent** (68.2), **Cortland** (60.6), *PRI 1661–2* (47.6), **Liberty** (34.8), **Jonamac** (31.7), **Fantazja** (24.9), <u>**Enterprise**</u> (14.4) |
| $A$-$Ma3$ (Granny Smith) | **Granny Smith**\* (62.0), *Lady Williams*, **Cripps Red** (47.5), <u>**Cripps Pink**</u> (43.4), Frostbite, Keepsake, Sweet 16 (11.9) |
| $B_{Mc}$-$Ma3$ (McIntosh) | **McIntosh**\***, Regent** (62.0), Goodland (56.9), **Cortland** (47.5), <u>**Enterprise**</u>, **PRI 1661–2** (43.7) |
| $C_{ES}$-$Ma3$ (Esopus Spitzenburg) | **Esopus Spitzenburg**\* (62.0), *Idared*, *Jonathan*, <u>**Arlet**</u>, **Autumn Crisp**, **Burgundy**, **Jonafree**, **Monroe**, **NY 752** (61.3), **Sawa** (34.3) |
| $C_{F2}$-$Ma3$ (F$_2$26829-2-2) | **F$_2$26829-2-2**\* (62.0), Ben Davis, Cortland (54.4), Early Cortland (50.5), **Dayton** (48.5), **Prima** (40.8), *PRI 14–126* (39.2), *Co-op 15*, <u>**WA 5**</u>, **PRI 1661–2** (26.9), *Liberty* (25.7), NY 65707–19 (22.0) |
| $D$-$Ma3$ (McIntosh) | **McIntosh**\*, **Empire**, **Jonamac**, **Macoun** (62.0), **Sunrise** (55.7), *NJ 90* (47.5), *Spartan*, <u>**WA 1**</u>, NY 65707–19 (35.2), **Fantazja** (29.2) |
| $E$-$Ma3$ (Grimes Golden) | **Grimes Golden**\*, *Golden Delicious*, <u>**Arlet**</u> (62.0), **Cameo** (48.6), **PRI 14–126** (41.8), **BC 8S-27-43**, **Gala**, **Nicola, Scired**, **Sciros, Sonya** (36.8), Delorgue (34.8), Granny Smith, Macoun, Liberty (13.8) |
| $B_{GG+DO}$-$ma3$ (Grimes Golden / Duchess of Oldenburg) | <u>**Honeycrisp**</u> (14.2) |
| $F$-$ma3$ (UP_Golden Delicious) | *Golden Delicious*, <u>**Aurora Golden Gala**</u>, <u>**Splendour**</u>, **WA 5**, **Ambrosia**, **Delblush**, **Ginger Gold**, **Cripps Red**, **Tsugaru** (62.0), **Blushing Golden** (59.3), **WA 2** (57.8), **NY 752** (55.7), **Autumn Crisp** (54.5), <u>**Cripps Pink**</u>, **Honeygold**, **Silken** (48.6), *Goldrush* (40.7), **Pinova** (40.3), <u>**WA 1**</u> (37.6), **Elstar** (35), **Sunrise** (30.2) |

(*Continued*)

**Table 2.** (Continued)

| Allele (ancestral source) | Cultivars with shared haplotypes at the locus (length of extended shared haplotypes in cM) |
|---|---|
| *G-ma3* (UP_Delicious) | **Delicious, Ambrosia, NJ 90, Spartan** (62.0), **Empire** (60.8), <u>**Aurora Golden Gala**</u> (57.3), *Gala*, *Kidd's Orange Red*, **Scifresh** (55.8), **Fuji** (51.6), **Cameo** (48.6), <u>**Splendour**</u>, **Braeburn, Chinook, Nicola, Scired, Sciros, Sonya, WA 2** (45.2), **NY 543** (40.3), **Chinook** (38.4), Wagener, Idared, Fiesta (36.1), Cox's Orange Pippin, Clivia (21.3), NY 543 (18.3), Lodi, Ginger Gold, Montgomery (17.1), Fortune (15.2), Fireside, Minnewashta (14.2) |
| *H-ma3* (Winesap) | **Winesap**\* (62.0), <u>**Delicious, Melrose**</u> (60.1), Co-op 17 (38.5), Fortune (22.8) |
| *I-ma3* (UP_Jonathan) | *Jonathan*, **Akane, Sansa, Tsugaru, Wealthy, Fireside** (62.0), **Empress** (51.0), **Jonamac** (49.7), **Oriole** (44.5), <u>**Enterprise**</u> (25.7) |
| *J-ma3* (Northern Spy) | **Northern Spy**\* (60.2), <u>*Keepsake*</u> (60.8), **Jonafree** (34.8), <u>**Honeycrisp**</u> (31.7), James Grieve (17.5), Cox's Orange Pippin, Kidd's Orange Red, Fiesta, Ingrid Marie (15.6) |

Individuals in **bold** shared the haplotypes via IBD, while the rest shared haplotypes via IBS. <u>Underlined</u> individuals are the nine important breeding parents of [40]. *Italicized* individuals are ancestors of the important breeding parents for which allele assignment was also determined in [40]. Individuals annotated with an asterisk (\*) are ancestral sources of alleles. Immediately flanking haplotypes of the trait loci were identical for the following sets of alleles: $B_{F2/J}$-*Ma*, $B_{WS}$-*Ma*, and *G-Ma*; *H-ma* and *I-ma*; $C_{F2}$-*Ma3* and $C_{ES}$-*Ma3*.

enabled assignment of the second locus allele to the other flanking haplotype pattern present. If it was not possible to assign all locus alleles to each haplotype pattern at this point, DNA-profiled cultivars with known close pedigree connections and historical allele information were used to help identify shared homologs and thereby which locus allele should be assigned to which haplotype pattern. In cases where multiple alleles of a target locus were associated with a single flanking haplotype pattern, haplotypes of additional upstream and downstream haploblocks were considered one haploblock at a time until multi-haploblock haplotype patterns were uniquely associated with each functional allele (Fig 1). When adding these additional SNP haploblocks, the target locus was kept at the center of the multi-haploblock cluster and adding to the nearest new haploblock first. Then, additional haploblocks were considered progressively on each side so that the included flanking haploblocks downstream and upstream of the target locus covered a near-equal genetic length.

Once all alleles for target each locus had been associated with one unique SNP haplotype pattern in the reference panel individuals with their historically recorded locus genotypes (i.e., a single SNP haplotype pattern was not associated with more than one target locus allele), the ancestral source of each SNP haplotype pattern was determined. Using pedigree information, SNP haplotype patterns of individuals were compared to those of their progenitors to identify inheritance pathways. The earliest known ancestor having each SNP haplotype pattern was considered the ancestral source for that SNP haplotype pattern. Next, the haplotype patterns present in all cultivars and selections in each Crop Reference Set with unknown genotypes for each target locus were compared to the SNP haplotype patterns of the ancestral sources to identify all cases of haplotype sharing (Fig 2). Where an individual with an unknown target locus genotype shared its haplotype pattern with an ancestral source, the locus allele of the source associated with the SNP haplotype pattern was assigned to the individual. Where the inheritance of SNP haplotype patterns could be traced via known pedigree connections to a shared ancestor, the target locus allele was noted to be deduced via identity-by-descent (IBD). For cultivars with newly assigned alleles that could not be traced through known pedigree

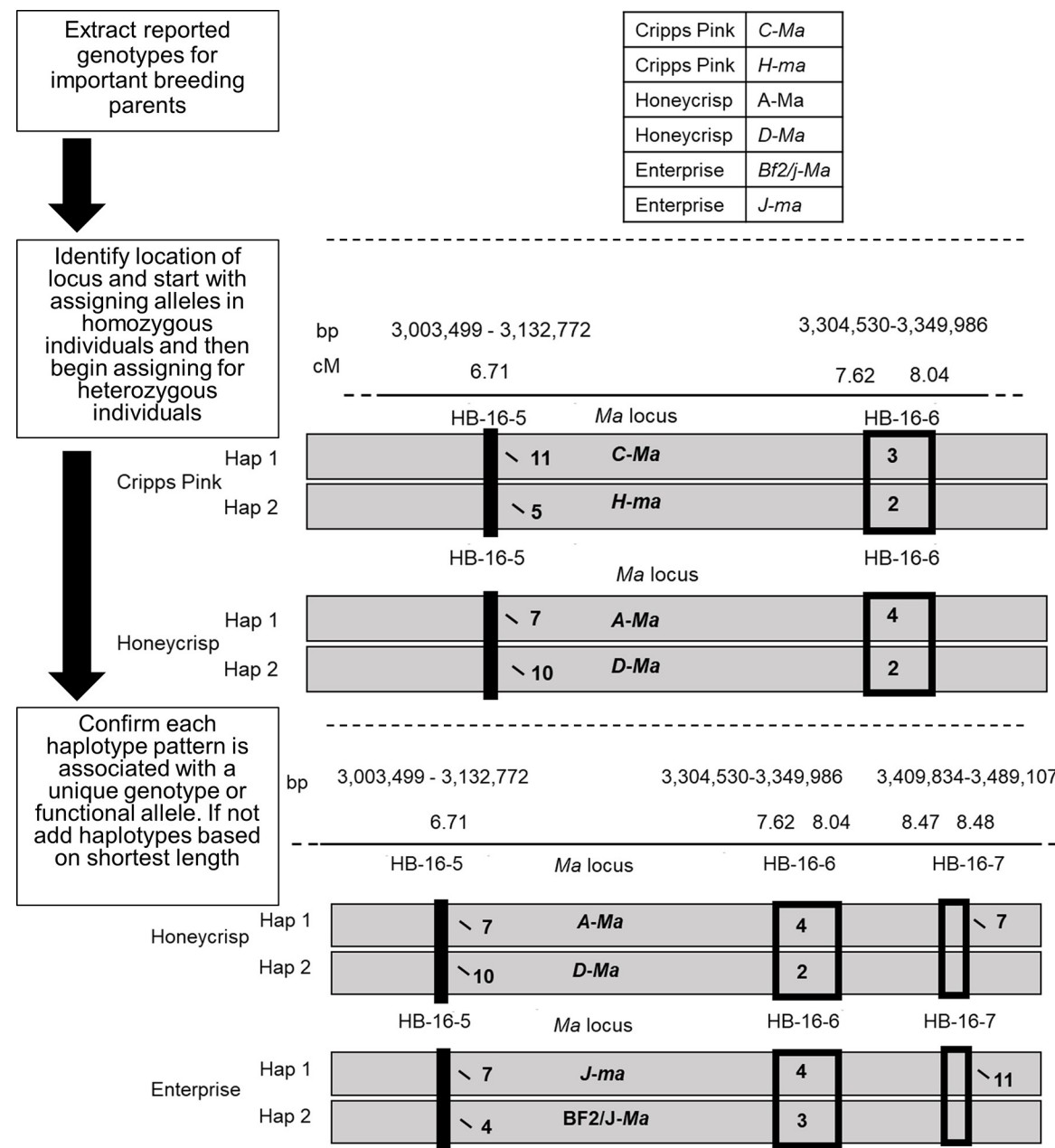

**Fig 1. Workflow for assigning alleles.** The workflow utilized to assign locus alleles from reference individuals (nine important breeding parents and founders) to unique flanking haplotype patterns from SNP array data.

connections to ancestral sources using SNP haplotypes, target locus alleles were noted to be deduced via identity-by-state (IBS).

The total length of extended haplotype sharing with ancestral sources across adjacent haploblocks to the trait locus was also recorded. In cases where the same flanking haplotypes were identical to more than one ancestral source for the same allele, alleles were assigned according to IBD if possible or else according to the allele of the ancestral source with the longest extended shared haplotype. While IBD segments could be tracked through the pedigree for high confidence in identity, there was less certainty about IBS segments being truly identical between individuals, especially for short segments. Therefore, alleles assigned via IBS were

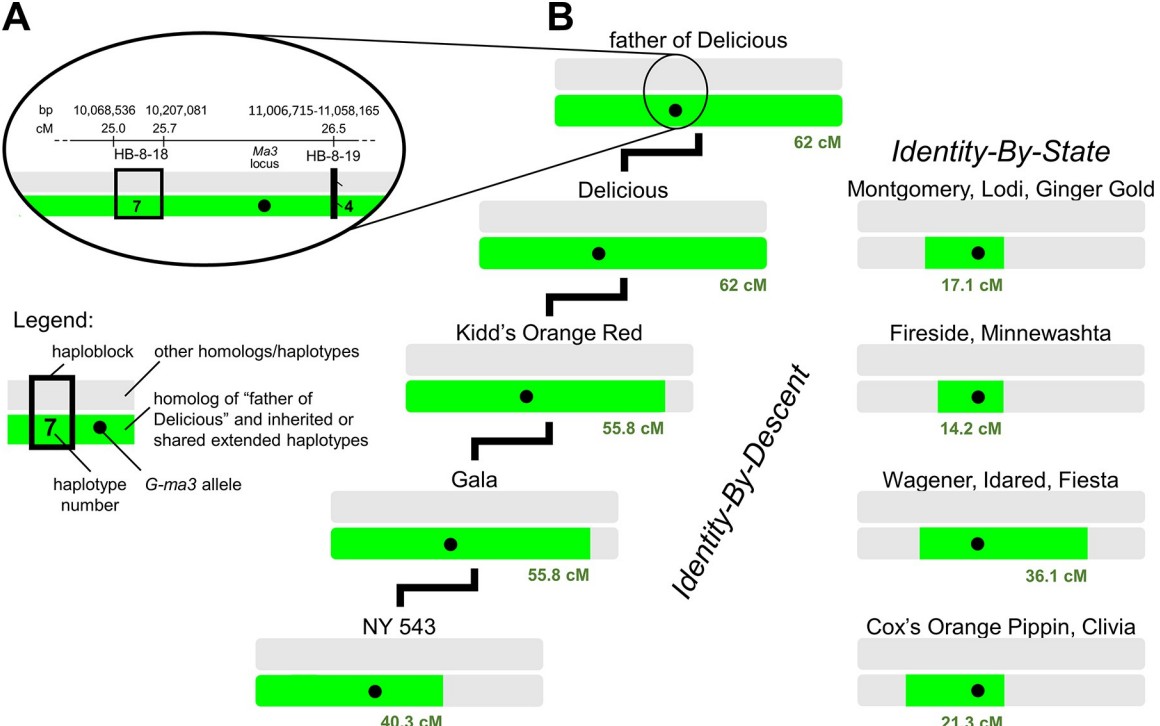

**Fig 2. Allele deduction for IBD and IBS cases.** Allele deduction via IBD (identity-by-descent) or IBS (identity-by-state) to tracking shared haplotypes in which an allele of interest is embedded, exemplified by the *G-ma3* allele of the father of 'Delicious'. Shown in green are extended haplotypes in coupling-phase linkage with *G-ma3* that are shared with the father of 'Delicious' without disruption by recombination; all other haplotypes are in gray. (**A**) The position of the *Ma3* locus is shown relative to haploblocks of chromosome 8 (only the immediately flanking haploblocks are shown). The exact position of *Ma3* is indicated using the physical position of the informative SNP identified in [43] and the flanking haploblocks are encompassed in the QTL regions identified in [40, 43]. The *G-ma3* allele, shown as a black dot, is flanked by haplotypes 7 and 4 of HB-8-18 and HB-8-19, respectively. (**B**) Extended haplotypes in which the *G-ma3* allele is embedded that are shared with a particular ancestor (father of 'Delicious') via IBD or IBS are shown for various cultivars. The entire length of chromosome 8, horizontal bars, is displayed for all individuals.

only listed and considered successfully deduced if they had a longer extended shared haplotypes than the shortest extended shared haplotype observed for IBD segments in the data set, which was 9.4 cM.

## Validation of deduced alleles

Locus genotypes of an additional 28 and 43 individuals, not previously used to associate haplotype alleles with locus alleles, were extracted from [4] for $R_f$ and GRIN-Global for *GD12*, respectively. In addition, 49 individuals were independently genotyped for *GD12* as follows: DNA for each individual was extracted according to [56], the GD12 SSR was amplified using primers and PCR conditions described in [6], resulting amplicons were separated and detected with an Applied Biosystems® 3730 DNA Analyzer, and observed amplicons were scored using GeneMarker® software. The proportion of new genotypic data for *GD12* that matched allele deductions of each individual was calculated as the accuracy of deduction. Mismatches were examined carefully to determine whether they were due to incorrect genotype calls in the GRIN-Global data set or real mismatches between deductions from SNP data and observations from marker genotyping. It was not possible to validate the results of *Ma* and *Ma3* because allele designations were based on QTL analyses and such analyses, or the data required to conduct such analyses, were not available for any of the individuals with deduced genotypes.

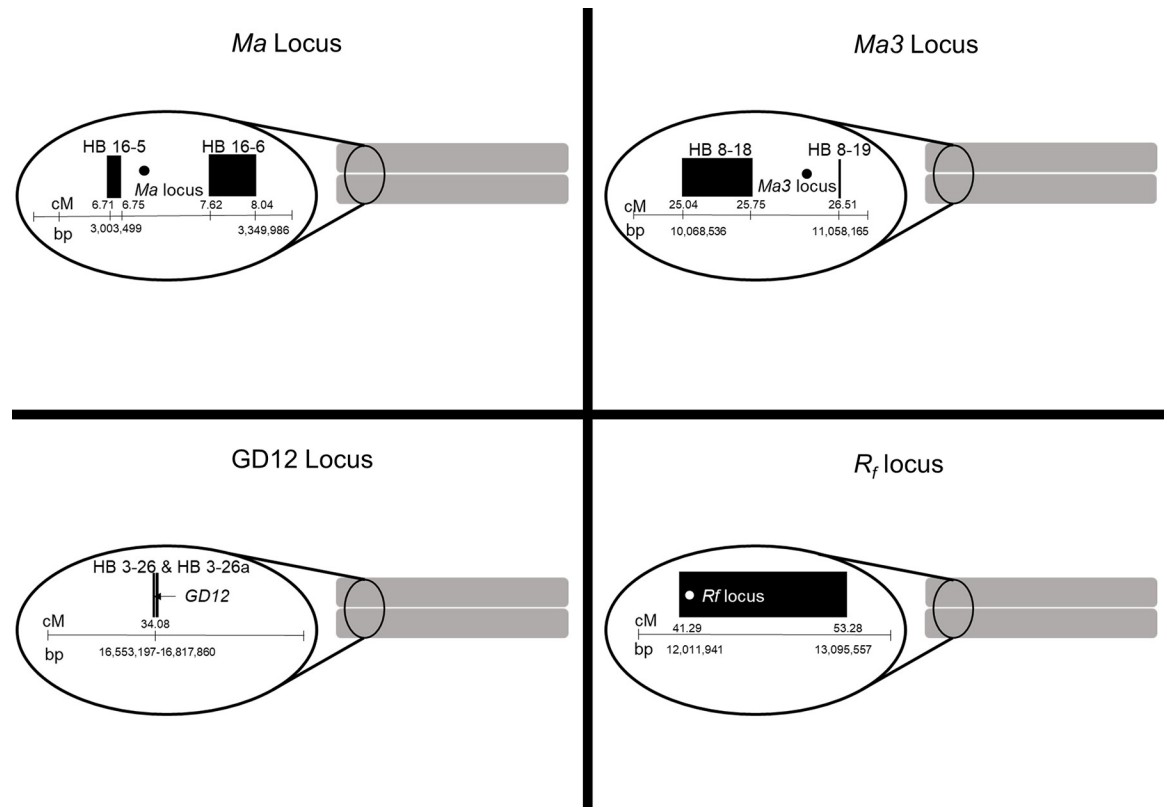

**Fig 3. Location of each locus in relationship to flanking haploblocks.** Upper left is the *Ma* locus in apple, upper right is the *Ma3* locus in apple, lower left is the *GD12* locus in apple, and lower right is the *Rf* locus in sweet cherry.

## Results

### Locus genomic positions

The physical position of the apple *Ma* locus (chromosome 16, 3177899 bp) was determined to be between HB-16-5 (3003499–3132772 bp) and HB-16-6 (3304530–3349986 bp).*Ma3* (chromosome 8, 10857522bp) was determined to be between HB-8-18 (10068536 - 10207081bp) and HB-8-19 (11006715–11058165 bp), which was situated at the end of the consensus QTL positions determined in [40, 43] (Fig 3). The sweet cherry *Rf* locus (chromosome 3, 121083592–12083916 bp) was determined to be within HB-3-17 (12403758–13095557) (Fig 3). The physical position of the apple *GD12* locus (chromosome 3, 16649441–16649311bp) was determined to be between HB-3-26 (16553197–16553304 bp) and HB-3-26a (16817860 bp) (Fig 3).

### Allele deduction and validation for QTLs

Successful deduction of alleles via both IBD and IBS of extended shared haplotypes with ancestral sources of the reference panel of nine important breeding parents (Arlet, Aurora Golden Gala, Cripps Pink, Delicious, Enterprise, Honeycrisp, Splendour, WA 5, W1) was achieved for a high proportion of individuals of both apple QTLs. In total, at least one allele was deduced for 64% and 73% of the Crop Reference Set cultivars and selections for *Ma* and *Ma3*, respectively (S2 Table). Complete genotypes (two alleles) for both the *Ma* and *Ma3* loci were deduced for 16 cultivars (14% of the 113 cultivars, excluding the nine important breeding parents), and

at least one allele of both loci was deduced for a further 49 cultivars (43%). For the *Ma* locus alone, complete genotypes were deduced for 23 cultivars (20%) and one allele for 49 cultivars (42%). At the *Ma* locus, 70 homologs matched via IBD and 25 homologs via IBS, for which the IBS threshold was established as ≥9.4 cM (Table 1). For the *Ma3* locus, complete genotypes were deduced for 38 cultivars (34%) and one allele for 44 cultivars (39%). At the *Ma3* locus, 91 homologs matched via IBD and 29 homologs via IBS (Table 1). No alleles for *Ma* and *Ma3* could be deduced for 26 individuals that had no pedigree connection and also did not share extended haplotypes with the nine important breeding parents. Among the individuals with missing allele information for *Ma*, 24 unique haplotypes patterns were observed, with just six of these accounting for 76% of the undeduced allele cases. *Ma3* had 21 unique haplotypes not assigned to a known functional allele, with just three of these representing 58% of undeduced allele cases.

## Allele deduction and validation for an MTL

Both alleles of $R_f$ were deduced for 45 (75%) of the 60 sweet cherry cultivars and selections (including the 16 ancestral sources) and one allele for an additional seven individuals (12%), resulting in at least one deduced allele for 86% selections of the sweet cherry Crop Reference Set (S2 Table). Via IBD, 86 homologs matched, while 11 homologs matched via IBS (S3 Table). No alleles could be deduced for eight individuals due to missing haplotype data (five) or unique haplotype patterns (three). For the homologs that could not be deduced, 10 unique haplotype patterns were detected, however none were common. All deduced alleles matched genotypes reported by [4], resulting in a 100% deduction accuracy for this locus.

## Allele deduction and validation for an SSR

For *GD12*, both alleles were deduced for 81 (67%) of the 121 cultivars and one allele for 28 cultivars (24%; S2 Table). Among deduced alleles, 167 were deduced via IBD and 23 via IBS (S4 Table). It was not possible to deduce any alleles for 12 individuals of the Crop Reference Set because they were not pedigree-connected to others and their haplotypes did not match via IBS to the ancestral sources. For the undeduced alleles of *GD12*, 31 unique haplotype patterns were observed with seven of those patterns being present in more than one of the undeduced individuals. A total of 93 deduced alleles were validated using newly obtained SSR data for 49 individuals (95% of alleles present). Of the remaining five alleles, four could not be validated due to poor DNA quality of two individuals and resulting lack of PCR amplicons, while the last allele was associated with a unique haplotype pattern. Thus, all allele deductions that could be validated via independent and de novo genotyping were correct.

For validation of 77 allele deductions with GRIN-Global data, three deduced alleles (4%) did not match the reported alleles, each occurring in a separate cultivar (Arlet, Early Cortland, and Worcester Pearmain). Further comparison of the reference alleles, deduced alleles, and extended haplotypes of these three cultivars with those of their parents, siblings, and offspring indicated that alleles were likely deduced correctly but that the GRIN-Global data contained errors (S5 Table). 'Arlet' was deduced as "155":"195" but reported as "155":"155". Its parents were reported as "155":"195" ('Golden Delicious') and "155":"155" ('Idared'), thus making both genotypes possible. However, one homolog of 'Arlet' matched the 'Golden Delicious' "195"-containing homolog across the entirety of chromosome 3. Thus, it was deduced that 'Arlet' should be "155":"195". 'Early Cortland' was deduced as "155":"187" but reported as "155":"155". Its parents were reported as "155":"187" ('Cortland') and "155":"187" ('Lodi'). However, one homolog of 'Early Cortland' matched the "187"-containing homolog of 'Cortland', inherited in turn from 'McIntosh', for the entirety of the chromosome and shared 48.5

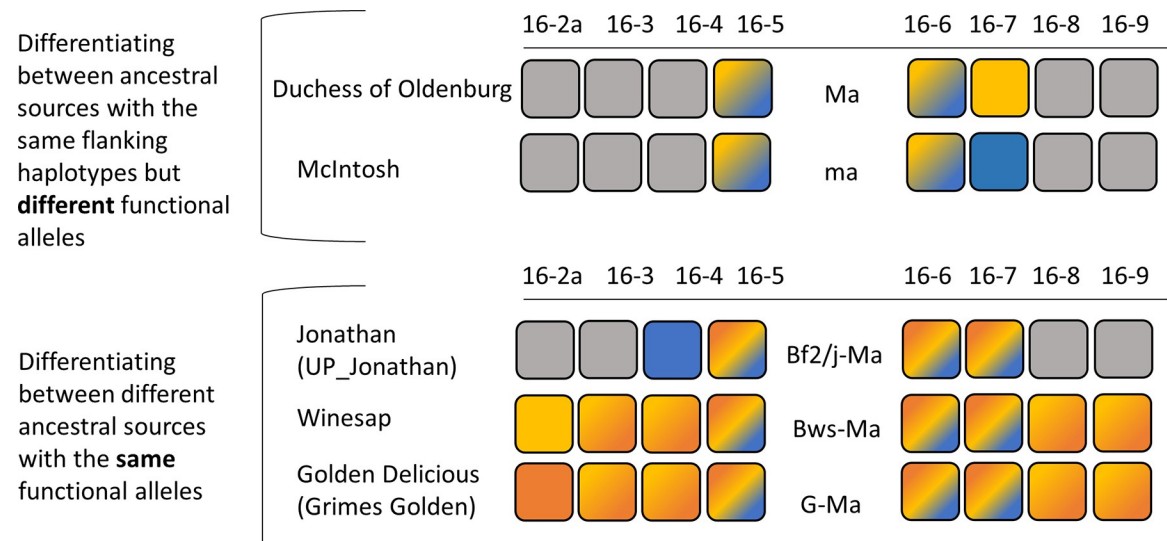

**Fig 4. Example of cases with IBS between functional alleles and among ancestral sources.** The first example compares a homolog of 'McIntosh' with that of 'Duchess of Oldenburg' inherited by 'Honeycrisp'. In this case, the homologs share the same haplotype pattern in the haploblocks immediately flanking the locus (HBs 16–5 and 16–6) but were reported to be associated with different functional alleles (*Ma* vs. *ma*). To differentiate between the two functional alleles, haplotypes of adjacent haploblocks were needed from haploblocks 16–5 to 16–7, spanning a total of 1.8 cM across the locus. The second example compares the homolog of 'Jonathan' transmitted by its other parent than 'Esopus Spitzenburg', the homolog of 'Winesap' inherited by 'Aurora Golden Gala', 'Delicious', and 'Splendour', and the homolog of 'Golden Delicious' transmitted by 'Grimes Golden'. In this example, all three have the same functional allele (*Ma*). However, to differentiate among the ancestral sources it was necessary to extend to adjacent haploblocks. The nearest haploblock was considered first and then the closest additional flanking haploblocks included one at a time either side of the locus until a unique extended patterns were identified. In this case, it was necessary to extend from haploblocks 16–2a to 16–9, spanning a total of 8.6 cM across the locus. Haplotypes shown with the gradient were those shared between individuals, while those in solid colors were those needed to be included to differentiate among ancestral sources that otherwise had the same immediate locus-flanking haplotypes but associated with different functional alleles or with different ancestral sources and the same functional allele.

cM across the GD12 locus with 'McIntosh'. 'Early Cortland' also shared 38–48.5 cM across this locus with other individuals that had inherited the 187-containing homolog from 'McIntosh'. Thus, it was deduced that Early Cortland should be "155":"187". 'Worcester Pearmain' was deduced as "155":"155" but reported as "155":"187". Although no parental information was available for this cultivar, its offspring had validated alleles of "155":"155" ('Discovery') and "155":"155" ('Lord Lambourne') and both individuals were determined to have inherited two different 'Worcester Pearmain' homologs. Thus, it was deduced that Worcester Pearmain should be "155":"155".

## Identity-by-state among ancestral sources

Both QTLs had cases of ancestral IBS for the same functional allele (Fig 4 & S6 Table). For the *Ma* locus, cases of haplotype-sharing between two sources with different functional alleles occurred with the *A-Ma/J-ma* alleles. These ancestral sources only shared the immediate flanking haplotypes, so it was possible to differentiate between functional alleles by considering one additional haploblock on either side. All other functional alleles could be differentiated with just the two flanking haplotypes. In order to differentiate between ancestral sources sharing the same functional allele, it was necessary to include up to four flanking haploblocks on each side for a 8.6 cM haplotype pattern across the locus (S6 Table). For the *Ma3* locus, there were two cases where it was necessary to distinguish between different functional alleles with the same flanking haplotypes. The first case was between $B_{Mc}$-*Ma3*, *E-Ma3*, $B_{GG+DO}$-*ma3*, and *J-ma3* and the second case was between *D-Ma3*, *F-ma3*, and *H-ma3*. For the first case, up to five

flanking haploblocks were needed on either side for a 12.7 cM haplotype pattern across the locus. The second case needed up to two flanking haplotypes on either side for 5.5 cM across the locus. All other functional alleles could be distinguished by just the flanking haplotypes. To differentiate between all ancestral sources sharing the same functional allele and the same haplotypes immediately around the locus (although they might represent an IBD allele just beyond the known pedigree), up to eight adjacent haploblocks on either side were needed, totaling up to 16.4 cM across the locus (S6 Table).

For the MTL, there was one case (haplotype 6) for which it was necessary to include flanking haploblocks to distinguish between different functional alleles. Inclusion of both flanking haploblocks on each side provided 14.4 cM of extended haplotypes that fully distinguished between the $R_f$ and $r_f$ alleles (S6 Table). In all other cases for the MTL, it was only necessary to include the haploblock in which the locus was embedded. To effectively differentiate among ancestral sources with the same functional allele, up to seven haploblocks on each side of the locus were needed, for up to 36.3 cM in total across the locus (S6 Table). The first case involved six individuals ('Ambrunes', 'Bertiolle', 'Emperor Francis', 'Empress Eugenie', 'Napoleon', and 'Schmidt') that all had haplotype 2, associated with the recessive $r_f$ allele. All shared 1–7 haplotypes on either side of the locus, totaling 14.2–36.3 cM across the $R_f$ locus. In the second case, both individuals (MIM 17 and MIM 23) had haplotype 18, associated with $r_f$ and shared seven flanking haplotypes on either side of the locus totaling 29.9 cM across the $R_f$ locus. In the third case, 'Summit' and 'Schmidt' shared six flanking haplotypes on either side of the locus (28.1 cM), with haplotype 23 associated with the dominant $R_f$ allele. The fourth case was between 'Blackheart' and PMR-1, which had haplotype 5 associated with $R_f$ and shared seven flanking haplotypes on either side of the locus (36.3 cM). The fifth case was between 'Ambrunes', 'Bertiolle', and 'Cristobalina', all of which had haplotype 8 associated with $R_f$ and shared one or two flanking haplotypes on either side of the locus (14.2–16.1 cM) (S6 Table).

For the GD12 SSR locus, to differentiate among all functional alleles from different ancestral sources, up to three flanking haploblocks on either side of the locus (4.5 cM) were needed (S6 Table). There were four cases of IBS among multiple ancestral sources (S6 Table). The first case involved the "155" allele that was shared by 'Beauty of Bath', 'Cox's Orange Pippin', 'Esopus Spitzenburg', 'Granny Smith', 'Malinda', 'McIntosh', both homologs of 'Northern Spy', 'Wagener', both homologs of 'Worcester Pearmain', and the unknown parent of 'Golden Delicious'. These nine ancestral sources shared 2–22 haploblocks (2.6–54.5 cM) across the GD12 locus. This "155" allele was the most common in the germplasm, with 59 additional cultivars having the allele and six of them matching via IBS. The second case of a shared haplotype was for the "157" allele of 'Beauty of Bath', 'Esopus Spitzenburg', 'Montgomery', 'Rome Beauty', and 'Russian Seedling'. These ancestral sources shared up to 9.8 cM across the locus, so it was necessary to expand four haploblocks on both sides of the locus to differentiate among them. The third case was the "159" allele of 'Ben Davis', 'Cox's Orange Pippin', 'Granny Smith', 'Malinda', 'Winesap', and the father of Delicious' (UP_Delicious). These individuals shared the same flanking haplotype pattern at the locus, so it was necessary to include 9–22 haploblocks on both sides of the locus (15.29–54.5 cM) to differentiate them all. The fourth case was the "187" allele of 'McIntosh and 'Montgomery' for which 2.9 cM was shared across the locus, so it was necessary to include three additional haploblocks on both sides to differentiate these ancestral sources (S6 Table).

## Discussion

We successfully developed, demonstrated, and validated a method to deduce alleles from SNP array data for various types of loci that extrapolates known allele information for a few

individuals to a larger germplasm set. In all cases where alleles could be deduced via IBD (i.e., for which inheritance of haplotypes could be traced from a shared ancestor), allele assignments were made with higher confidence than via IBS. While the method was demonstrated in apple and sweet cherry using the locus types represented by *Ma*, *Ma3*, *$R_f$*, and *GD12*, it could be expanded to other loci, other types of loci, and other crops that have SNP arrays or other genome-wide data available, especially where linkage disequilibrium is high. This approach enables germplasm users to extract information from previously characterized loci as well as newly developed assays not incorporated in a SNP array and extend this information to further individuals genotyped with the SNP array.

The developed method could be used to confirm reported genotypes. SSR genotyping is not always accurate as was identified here and has been reported in other studies [57–59]. Confirmation of reported results is important to ensure accuracy of published allele information for individuals. In all three cases where the GD12 allele did not match the GRIN-Global data, there were other validated individuals that had extended haplotypes matching the individual across the locus. While it is possible that a double recombination occurred at the location of the GD12 locus, these are rare events and highly unlikely to occur in the same genomic position in all three individuals. Alternatively, while parent-child and parent-parent-child errors (also called Mendelian-inconsistent errors) can be detected relatively easily, Mendelian-consistent errors (genotyping errors that do not infringe on Mendel's inheritance laws) are harder to detect and require the phasing of linked loci [46]. Although no Mendelian-inconsistent errors were observed in the GRIN-Global data set, it is unlikely that any Mendelian-consistent errors were detected and resolved, especially because no or few flanking markers were available to conduct such error removal. Thus, it is more likely that the GRIN-Global data was incorrect as there were no possibilities for correction of Mendelian-consistent errors. Application of the method here easily identified the genotypic errors and could be systematically performed for listed genotypes of other loci in GRIN-Global datasets or reported elsewhere.

Cases of IBS among ancestral sources were detected for all loci investigated in this study. For IBS among ancestral sources with different functional alleles, the identical segments were often very short, with the longest being 14.5 cM (certain individuals with haplotype 6 of the *$R_f$* locus of cherry). However, to differentiate among ancestral sources with the same functional allele, it was often necessary to examine extended haplotypes on one or both sides of the target loci. Recent studies have reported that many historic apple cultivars are closely related with unknown recent shared ancestors [27, 60]. Thus, while these extended haplotype patterns with identical functional alleles were treated as originating different ancestral sources in this study, it is likely that in many of these cases a shared recent ancestral is the source of the allele. Therefore, both the IBD and IBS deductions capitalized on a high degree of linkage disequilibrium among the cultivated germplasm.

The many haplotypes observed in both apple and cherry that were not able to be associated with known alleles via IBD or IBS present opportunities for further research. While most of these allele-unassigned haplotypes were from individuals not pedigree-connected with other germplasm or poorly represented in the germplasm, there were also cases of haplotypes present in common ancestors but not represented in previous QTL studies. For example, the second *Ma* allele of the ancestor 'McIntosh´ had extended haplotype-sharing via IBD or IBS with eight other cultivars but was not functionally characterized in the multi-parent study [40]. Therefore, its association with high or low acidity is unclear. To ascertain allele effects, an efficient approach would be to conduct DNA testing, or ideally QTL analyses, for sets of individuals representing the most common undetermined haplotypes (highlighted in S5 Table). The method established in this study could then be applied to quickly deduce allele identities for all individuals sharing those haplotypes, efficiently expanding the number and proportion of

germplasm individuals with genotypic information for loci of interest. Thus, the availability of a reference data set covering all or most of the observed haplotypes in relevant germplasm would be of much value for confident germplasm usage.

Opportunities for improvement of this method include determining extended haplotypes that are unambiguously associated with each allele as well as extending the method to unphased SNP data. Alleles deduced via IBD in this study were deduced unambiguously because pedigrees of these individuals were known, an approach originally outlined in [61], enabling establishment of IBD relationships for chromosomal regions among individuals. The ability to unambiguously assign alleles for loci where pedigree connections are unknown would greatly expand the allele information available. To do so, additional diagnostic SNPs could be developed and specifically used to genotype key individuals, or these additional SNPs could be included in future genome-wide assays. However, for immediate use of genotypic data sets in which ambiguity persists, some efficient shortcuts are available. Establishing thresholds of shared haplotype lengths by empirically determining the lengths at which matching of a known allele is unambiguous would enable rapid and confident allele assignment in IBS cases. Here, ≥9.4 cM was used as the threshold, taken from the minimum shared length observed via IBD with an ancestral source (which allowed for recombination to shorten shared haplotypes) among all the examined loci. Other methods for establishing confidence of deductions could be used, relying on empirical observations or theoretical calculations. For individuals with shared haplotypes that are not above the thresholds of unambiguity, alleles could be assigned according to their longest match, with the degree of confidence assigned according to the previously described empirical observations. Additionally, expanding the approach to unphased data could enable rapid extraction of valuable information from genome-wide SNP assays (such as SNP arrays or genotyping-by-sequencing), bypassing the time and effort for the data curation step of phasing, although at the expected cost of some loss of accuracy. Ultimately, the automation of such a method could enable genome-wide SNP data to be rapidly interpreted into allele information simultaneously for any and many loci, instead of obtaining information from one DNA test or genetic marker at a time. A streamlined process would further increase the ability for germplasm users to quickly gain allelic information about loci of interest for their germplasm while providing increased confidence in the utilization of genetic resources.

## Supporting information

**S1 Table. SNPs included in the study.** Details are provided on each SNP's name, NCBI dbSNP accession identifier, linkage group and genetic position, haploblock, and chromosome and physical position. For apple, details were extracted from Vanderzande et al. (2019)*. For sweet cherry, SNP name and identifier and physical chromosome and position were extracted from Vanderzande et al. (2020); genetic position and haploblock were extracted from Vanderzande et al. (2019)*. *Dataset available at https://www.rosaceae.org/publication_datasets, accession number tfGDR1038.
(XLSX)

**S2 Table. All alleles deduced for four loci utilized in this study for apple and cherry.**
(XLSX)

**S3 Table. Alleles deduced for the $R_f$ locus for sweet cherry cultivars and selections by haplotype sharing.** Sharing via IBD is shown in **bold**, otherwise sharing was via IBS. Individuals annotated with an asterisk (*) are ancestral sources of alleles. For H13, 'Windsor' and 'Venus' shared the same extended haplotypes with 'Blackheart' and 'PMR-1', so they are listed under

H13 for both ancestral sources.
(DOCX)

**S4 Table. Alleles deduced for the GD12 locus for various apple cultivars and selections by haplotype sharing.** Sharing via IBD is shown in **bold**, otherwise sharing was via IBS. Individuals annotated with an asterisk (*) are ancestral sources of alleles.
(DOCX)

**S5 Table. Haplotype comparisons for three cultivars with alleles deduced for the SSR GD12 that did not match reported genotypes on GRIN-Global.** As evidence of correct deduction, extended haplotype patterns are shown for the cultivars and their parents, some siblings, and some offspring. Extended haplotype patterns are color-coded by ancestral source.
(XLSX)

**S6 Table. Display of extended haplotypes of ancestral sources needed to differentiate among all functional alleles.** Flanking haplotypes with the same background shades have the same pattern. Cells with green fill represent the locus and its immediately flanking haplotypes, extended haplotypes with gray fill were necessary for differentiation among functional alleles, those in blue fill were necessary for differentiation among ancestral sources. Cells with yellow fill were the haploblock necessary to differentiate that ancestral cultivar from others that had the same flanking haploblock but different functional alleles, cells with red fill were the haploblock necessary to differentiate between ancestral cultivars that had the same flanking haploblocks and the same functional alleles.
(XLSX)

## Acknowledgments

Jack Klipfel's assistance with conducting the SSR genotyping is gratefully acknowledged.

## Author Contributions

**Conceptualization:** Cameron Peace.

**Data curation:** Stijn Vanderzande.

**Funding acquisition:** Cameron Peace.

**Investigation:** Alexander Schaller.

**Methodology:** Alexander Schaller, Stijn Vanderzande, Cameron Peace.

**Project administration:** Cameron Peace.

**Supervision:** Stijn Vanderzande, Cameron Peace.

**Validation:** Alexander Schaller.

**Visualization:** Alexander Schaller, Cameron Peace.

**Writing – original draft:** Alexander Schaller.

**Writing – review & editing:** Alexander Schaller, Stijn Vanderzande, Cameron Peace.

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
