## [Decision Letter · Decision Letter 0]

9 Sep 2022

PONE-D-22-21108Deducing genotypes for loci of interest from SNP array data via haplotype sharing, demonstrated for apple and cherryPLOS ONE

Dear Dr. Peace,

Thank you for submitting your manuscript to PLOS ONE. After careful consideration, we feel that it has merit but does not fully meet PLOS ONE’s publication criteria as it currently stands. Therefore, we invite you to submit a revised version of the manuscript that addresses the points raised during the review process.

We look forward to receiving your revised manuscript.

Kind regards,

Evangelia V. Avramidou, PhD

Academic Editor

PLOS ONE

Journal Requirements:

"This work was funded by USDA’s National Institute of Food and Agriculture-Specialty Crop Research Initiative Project “RosBREED: Combining disease resistance and horticultural quality in new rosaceous cultivars” (2014-51181-22378) and the USDA National Institute of Food and Agriculture Hatch project 1014919, Crop Improvement and Sustainable Production Systems (WSU reference 00011)."

"This study was supported by USDA's National Institute of Food and Agriculture-Specialty Crop Research Initiative project “RosBREED: Combining disease resistance and horticultural quality in new rosaceous cultivars” (2014-51181-22378) and the USDA National Institute of Food Agriculture Hatch project 1014919, Crop Improvement and Sustainable Production Systems (WSU reference 00011) for CP, SV, and AS. The funders had no role in study design, data collection and analysis, decision to publish, or preparation of the manuscript."

4. Please remove all personal information, ensure that the data shared are in accordance with participant consent, and re-upload a fully anonymized data set. 

Additional Editor Comments:

Dear authors,

Based on reviewer's comments and expertise I would suggest to procceed to appropriate corrections that would further improve your manuscript in order to be published.

Reviewers' comments:

Reviewer's Responses to Questions

**Comments to the Author**

1. Is the manuscript technically sound, and do the data support the conclusions?

Reviewer #1: Yes

Reviewer #2: Yes

2. Has the statistical analysis been performed appropriately and rigorously? 

Reviewer #1: Yes

Reviewer #2: N/A

3. Have the authors made all data underlying the findings in their manuscript fully available?

Reviewer #1: Yes

Reviewer #2: Yes

4. Is the manuscript presented in an intelligible fashion and written in standard English?

Reviewer #1: Yes

Reviewer #2: Yes

5. Review Comments to the Author

Reviewer #1: In the manuscript "Deducing genotypes for loci of interest from SNP array data via haplotype sharing,

demonstrated for apple and cherry", the authors developed a method for deducing genotypes at loci of interest. The authors validated the method using apple and cherry with three locus type as a case of study: 1. QTL: Ma and Ma3 for acidity in apple; 2. microsatellite marker GD12; and 3. Mendelian trait locus Rf for sweet cherry fruit color. The method developed by the authors enabled interesting findings. All approaches used in this manuscript are appropriate and represent standard methods used in this type of study.

Just some minor comments:

1. Introduction: I suggest that the authors briefly comment in the introduction section the case of study used in this manuscript for each type of loci (Ma and Ma3 for acidity in apple; 2. microsatellite marker GD12; and 3. Mendelian trait locus Rf for sweet cherry fruit color). Although the authors included some information in the materials and methods section, it would be helpful to introduce the QTL, MTL and the multi-allelic microsatellite marker used in this manuscript.

2. Lines 216-219: I suggest that the authors include the physical position (bp) in the text of the manuscript

3. Line 223: The authors cited in the manuscript the paper that reported the nine important breeding parents. Ideally, they should also include the name of the breeding parents between parenthesis. “Successful deduction of alleles via both IBD and IBS of extended shared haplotypes with ancestral sources of the nine important breeding parents…”

Reviewer #2: The manuscript presents a method to deduce alleles linked to traits of interest from SNP array datasets. Application of the methodology is shown for three different types of loci in apple and sweet cherry.

Although the method applied appears sounds, I found the manuscript confusing in parts and hard to follow. At the current state, it would be difficult, in my opinion, to replicate such method in other crops/loci. I think that many concept, methods and results could be written in a simpler and more clear way, and I strongly suggest to make more use of figures, even as supplementary material. Additionally, it would probably be useful to summarize findings from cited articles that have been used as foundation for this work.

Below I highlighted some paragraph that I think could be improved:

- lines 77-83: this can be re-written a bit more clearly. I see you want to differentiate between high-density SNP arrays (which are mainly used for association mapping analysis and genomic predictions) and the need of trait-targeted and high-throughput genotyping systems for breeding and germplasm management.

- lines 83-91: again, I think this can be simplified a little. While newly developed SNP arrays can include validated markers linked to traits of interest, your objective here is to develop a method to identify samples carrying favorable alleles using SNP array-datasets already generated.

- lines 140-157: this part is rather confusing. Could you maybe re-write in a clearer way, and provide tables/figures showing the data used and the process applied?

- lines 163-165: does this mean that you used pedigree information to deduce the haplotype patterns? What does it meant "selections with unknown genotypes"? Didn't you use only accessions that had been genotyped with SNP arrays (as from paragraph "Data set")?

- line 198: what about validating with phenotypic data? If the objective is to identify samples carrying favorable alleles to certain traits, it seems important to verify if your method is correctly assigning the deduced alleles.

- line 215: maybe a figure here, with chromosome numbers, would be useful.

- lines 293-345: it is really hard to follow all the details reported in the results. I strongly suggest to make more use of figures to show your findings.

Minor revisions:

- line 67: change "tiny" with "small"

- line 70: change "numerous" with "several"

- line 79: change "transient breeding selections" with "seedlings at early stages of selection"

- line 80: I'd remove the sentence "tens of dollars (rather than cents to a few dollars)", since costs are in constant development

- lines 84-86: I find the sentence "Instead, genotypes for QTLs, Mendelian trait loci (MTLs), or any loci of interest such as multi-locus SSRs are not the immediate output of SNP arrays and are hidden within a sea of data points." a bit mis-leading, since SNP arrays are used often to identify loci of interest

- line 140: is this what is shown in Table S2?

- line 141: is this for a total of 121 accessions, as reported in previous paragraph?

- line 143: 60 cherry accessions?

6. PLOS authors have the option to publish the peer review history of their article (what does this mean?). If published, this will include your full peer review and any attached files.

Reviewer #1: No

Reviewer #2: No

---

## [Author Response · Author response to Decision Letter 0]

29 Dec 2022

The response to reviewers is detailed in a Word file included in the file submissions.

---

## [Decision Letter · Decision Letter 1]

24 Jan 2023

Deducing genotypes for loci of interest from SNP array data via haplotype sharing, demonstrated for apple and cherry

PONE-D-22-21108R1

Dear Dr. Peace,

We’re pleased to inform you that your manuscript has been judged scientifically suitable for publication and will be formally accepted for publication once it meets all outstanding technical requirements.

Kind regards,

Evangelia V. Avramidou, PhD

Academic Editor

PLOS ONE

Additional Editor Comments (optional):

Dear authors,

all the comments have been adressed and your manuscript is ready for publication.

With kind regards

Reviewers' comments:

Reviewer's Responses to Questions

**Comments to the Author**

1. If the authors have adequately addressed your comments raised in a previous round of review and you feel that this manuscript is now acceptable for publication, you may indicate that here to bypass the “Comments to the Author” section, enter your conflict of interest statement in the “Confidential to Editor” section, and submit your "Accept" recommendation.

Reviewer #1: All comments have been addressed

2. Is the manuscript technically sound, and do the data support the conclusions?

Reviewer #1: Yes

3. Has the statistical analysis been performed appropriately and rigorously? 

Reviewer #1: Yes

4. Have the authors made all data underlying the findings in their manuscript fully available?

Reviewer #1: Yes

5. Is the manuscript presented in an intelligible fashion and written in standard English?

Reviewer #1: Yes

6. Review Comments to the Author

Reviewer #1: The authors have satisfactorily addressed all my comments and made the necessary changes to the manuscript. Therefore, I am satisfied with the revision made on the manuscript and would like to recommend the

manuscript for publication.

7. PLOS authors have the option to publish the peer review history of their article (what does this mean?). If published, this will include your full peer review and any attached files.

Reviewer #1: No
